

# Double-target self-supervised clustering with multi-feature fusion for medical question texts

Xifeng Shen[1], Yuanyuan Sun[2], Chunxia Zhang[3], Cheng Yang[3], Yi Qin[1], Weining Zhang[1], Jiale Nan[1], Meiling Che[1] and Dongping Gao[1]

[1] Institute of Medical Information, Chinese Academy of Medical Sciences, Peking Union Medical College, Beijing, China
[2] National Center for Healthcare Quality Management in Rare Diseases, Virtual Human Platform, National Infrastructures for Translational Medicine, Institute of Clinical Medicine & Chinese Academy of Medical Sciences and Peking Union Medical College Hospital, Beijing, China
[3] School of Computer Science and Technology, Beijing Institute of Technology, Beijing, China

## ABSTRACT

**Background.** To make the question text represent more information and construct an end-to-end text clustering model, we propose a double-target self-supervised clustering with multi-feature fusion (MF-DSC) for texts which describe questions related to the medical field. Since medical question-and-answer data are unstructured texts and characterized by short characters and irregular language use, the features extracted by a single model cannot fully characterize the text content.

**Methods.** Firstly, word weights were obtained based on term frequency, and word vectors were generated according to lexical semantic information. Then we fused term frequency and lexical semantics to obtain weighted word vectors, which were used as input to the model for deep learning. Meanwhile, a self-attention mechanism was introduced to calculate the weight of each word in the question text, *i.e.*, the interactions between words. To learn fusing cross-document topic features and build an end-to-end text clustering model, two target functions, L cluster and L topic, were constructed and integrated to a unified clustering framework, which also helped to learn a friendly representation that facilitates text clustering. After that, we conducted comparison experiments with five other models to verify the effectiveness of MF-DSC.

**Results.** The MF-DSC outperformed other models in normalized mutual information (NMI), adjusted Rand indicator (ARI) average clustering accuracy (ACC) and F1 with 0.4346, 0.4934, 0.8649 and 0.5737, respectively.

Corresponding author
Dongping Gao,
gaodp_2022@126.com

## INTRODUCTION

In recent years, with the rapid growth of China's aging population, the allocation of medical resources has gradually been unable to meet the growing medical needs of the people, and the online consultation platform has received more and more attention, providing a platform and opportunity for patients and doctors to communicate remotely, and has accumulated a large number of medical question texts. However, the calculation

of the natural language processing method for the problem text can not achieve a high enough similarity, resulting in the clustering algorithm can not accurately cluster them into the same class cluster, and the calculation time of the model directly based on the neural network is long, if the patient's problem is matched with all the problems in the database, it takes a long time, the historical underlying problem data is clustered, and the similar questions are clustered into the same class cluster, which is essential for fully mining medical knowledge and building a high-quality medical intelligent question and answer system. Complementary medical care is of great significance.

The short text clustering is an important task which is challenging in many real-world applications. However, many short text clustering methods based on the bag-of-words (BoW) model lead to text representation sparsity, while neural network word embedding-based methods cannot capture document structure dependencies in text corpora. What's more, traditional text clustering algorithms compute text representation separately from the clustering process.

Self-supervised learning is a kind of unsupervised learning using unlabeled data. In this study, clustering results are used as a target self-supervised training for guiding the deep network to learn better representations. The deep network is trained jointly with clustering process to learn high quality text representations, which helps to improve the performance of the clustering algorithm including the clustering accuracy and the generalization ability of the network and changes the non-end-to-end nature between feature representation and clustering algorithms under the traditional clustering framework.

In medical question answering systems, unstructured text data plays a crucial role. These text data typically have short characters and irregular language usage, which makes feature extraction particularly complex. Since a single model is difficult to describe the text content comprehensively, this study proposes a multi feature fusion bi objective self supervised clustering (MF-DSC) method, which aims to build feature vectors with richer and more semantic information by combining multiple models. The main contributions of our work can be summarized as follows:

(1) A multi feature fusion (MF-DSC) strategy was proposed to extract features from texts describing medical problems. By combining term frequency and lexical semantics, we obtained a weighted word vector that can more accurately represent text content. This method not only considers the frequency of terminology appearing in the text, but also fully utilizes the semantic information of vocabulary, making the extracted features more representative.

(2) Introduced a self attention mechanism to calculate the weight of each word in the problem text. This mechanism can capture the interaction between words, thereby better understanding the semantics of the text. Through self attention mechanism, we can learn to integrate cross document thematic characteristics, making text clustering more accurate.

(3) This study also established an end-to-end text clustering model. This model constructs and integrates two objective functions–L cluster and L topic–into a unified clustering framework. This design helps to learn a friendly representation, thereby promoting text clustering. Through end-to-end training, the model can automatically

optimize feature extraction and clustering processes, improving the accuracy and efficiency of clustering.

# RELATED WORK

## Text representation methods

In the field of natural language processing (NLP), text representation is a crucial task. It involves transforming text into a form that computers can understand and process, enabling tasks such as sentiment analysis, topic classification, and information extraction to be performed. Among numerous text representation methods, bag of words (BoW) based methods and neural network-based methods dominate (*Soares et al., 2019*; *Kolari et al., 2006*; *Wu, Hoi & Yu, 2010*). The BoW based method is a simple and intuitive text representation method. It views text as a collection of words, without considering the order or grammatical structure between words. In this method, the text is represented as a vector, with each dimension of the vector corresponding to a word, and the value of that dimension represents the frequency or weight of that word appearing in the text. The main advantage of this method is that the generated text vectors are interpretable, as each dimension corresponds to a specific word, allowing for a clear understanding of the vector's meaning. In addition, the BoW method has shown good effectiveness in handling various text mining tasks.

However, with the increase of vocabulary, BoW based methods face some challenges. Firstly, the generated text vectors may become very high-dimensional and sparse. High dimensional vectors not only increase the complexity of computation and storage, but may also lead to the problem of "curse of dimensionality", where in high-dimensional space, most sample points are isolated from each other, rendering traditional distance measurement methods (such as Euclidean distance, cosine similarity, *etc.*) ineffective, thereby reducing clustering performance. Secondly, sparse vectors contain a large number of zero values, which do not contribute to text similarity calculation but occupy a large amount of storage space.

In order to overcome the limitations of the BoW based method, researchers have begun exploring neural network-based methods. Neural networks have powerful feature learning and representation capabilities, which can automatically extract useful information from text and encode it into low dimensional, dense vector representations. This method is commonly referred to as word embeddings or word vectors. Through the learning of neural networks, we can map semantically similar words to similar vector space positions, thereby achieving more effective text similarity calculation and clustering analysis.

In addition to neural network methods, there are also some other text representation methods, such as topic based representation methods (such as latent Dirichlet distribution LDA), and graph based representation methods (such as text graph models). These methods each have their own advantages and disadvantages, and are suitable for different NLP tasks and scenarios.

Considering these limitations including the high dimensionality and feature sparsity, some researches have focused on how to improve traditional normal text clustering

methods or models to analyze short texts. Collapsed Gibbs Sampling algorithm for the Dirichlet Multinomial Mixture model (GSDMM) (*Yin & Wang, 2014*) and feature dimensionality reduction including principal component analysis (PCA) (*Yan et al., 2012*) and non-negative matrix factorization (NMF) (*Huang, Zhou & Zhang, 2012*). The main idea of these methods is to analyze the relationship between term pairs to compensate for the sparsity of short texts and reduce the dimensionality of text vectors. However, these models assume the words in a given text are independent and ignore the information contained in the sequence order of words, which is also essential for understanding the text.

Recently, the neural network-based approach has provided new solutions, with significant improvements in many applications compared to BoW-based approaches. Due to the redundancy of natural language, models extract features from texts with variable length, such as phrases, sentences and texts. To capture the above features, neural network models are widely used as feature extractors. For text representation, Word2Vec (*Mikolov et al., 2013*) uses the BoW-based model or skip-gram model and is trained by hierarchical softmax or negative sampling to obtain word embedding. GloVe (*Pennington, Socher & Manning, 2014*) is a global log-bilinear regression model that makes full use of statistical information to train for word embedding. ElMo (*Peters et al., 2018*) applies a bi-directional language model to train a massive number of corpora for context-sensitive word embedding. The recurrent neural network (RNN) (*Hochreiter & Schmidhuber, 1997*; *Lee & Dernoncourt, 2016*) and convolutional neural network (CNN) (*Kim, 2014*) are widely used in text classification models as feature extractors to enrich the text feature (*Deng, Cheng & Wang, 2021*).

To counter the sparsity of text features, *Xu et al. (2015)* used pre-training and the CNN model to obtain the deep feature representation for short texts, and then used Kmeans for clustering. *Hadifar et al. (2019)* used SIF embedding to represent original short texts and perform self-supervised deep clustering. However, the sentence representations trained by the above methods encode semantic information only in sentences or their local contexts. Therefore, these representations cannot form high relevance within the same clustering category and may even form high similarity between different categories, thus misleading the clustering algorithm.

After analyzing BoW and neural network-based text representation methods, it is concluded that the existing models need to be improved due to sparse features, high-dimensional data, the neglect of the information contained in the word sequence, and the semantic information encoding of sentence representation solely in sentences or their local contexts.

## Self-supervised clustering

The basic idea of the self-supervised learning, a kind of unsupervised learning, is to design auxiliary tasks and build supervised information from unsupervised data to train, through which representations valuable for downstream tasks could be learned. Since it combines representation learning and clustering targets for deep clustering, the self-supervised learning changes the non-end-to-end nature of feature representation and

clustering algorithms under traditional clustering frameworks. Hence deep clustering algorithm researchers try to obtain a clustering-friendly representation by incorporating the clustering target function into representation learning network loss functions. *Yang et al. (2017)* studied K-means-friendly data representation using a K-means target function-assisted encoder. *Xie, Girshick & Farhadi (2016)* designed the KL-divergence loss function which makes the representation learning closer to the cluster centroid and thus improves the clustering results. *Guo et al. (2017a)* used a reconstruction loss function for deep clustering to constrain the auto-encoder, which helpd the auto-encoder to learn a better data representation. *Jiang et al. (2017)* achieved excellent clustering results by co-modeling the generative process with clustering using a deep variational auto-encoder. *Caron et al. (2018)* dealt with the clustering results as pseudo-labels to obtain better data representations for supervising the training of deep neural networks in large datasets. *Jin, Zhao & Ji (2020)* extracted topics from corpora by topic modeling, introduced cross-document information, and used document topics as auxiliary targets for short text clustering feature optimization.

To obtain a more friendly clustering representation, more studies have tried to fuse the clustering target function to the loss function of representation learning network. The loss of most deep clustering networks consists of two components, network loss $L_n$ and clustering loss $L_c$ (*Jia et al., 2018*).

According to the differences between network structure and loss functions, two training approaches can be obtained. One is to use $L_c$ as the only training loss function and only the clustering loss as the network training. Deep embedded clustering (DEC) (*Li, Qiao & Zhang, 2018*) simultaneously learns feature representations and cluster assignments using deep neural networks. It defines a distribution based on cluster centroids and regards minimizing clustering loss (KL divergence) as an auxiliary objective, making the pre-trained auto-encoder get a representation closer to the cluster centroid. It is a high confidence cluster distribution, which is better in terms of both clustering accuracy and speed. Discriminatively Boosted Clustering (DBC) (*Chen, 2015*), which is developed based on DEC, demonstrates a better performance on image dataset by convolutional auto-encoder. The other approach treats reconstruction loss as a kind of network loss function, and trains $L_n$ and $L_c$ jointly, as in Formula (1) Improved Deep Embedded Clustering (IDEC) (*Guo et al., 2017a*) and Deep clustering with convolutional autoencoders (DCED) (*Guo et al., 2017b*) have the same network structure as DEC, replace the convolutional auto-encoders with the original stack auto-encoders, and integrate the reconstruction loss and clustering loss into a unified framework, forming an end-to-end self-supervised model.

## METHOD

### Technical process

To improve the feature representation for texts and change the non-end-to-end nature between feature representation and clustering algorithm in the traditional clustering, this study proposes MF-DSC. Firstly, the weight of a word was obtained based on term frequency, word vectors were generated based on the semantic information, and term frequency is fused with word semantic information to obtain weighted word vectors

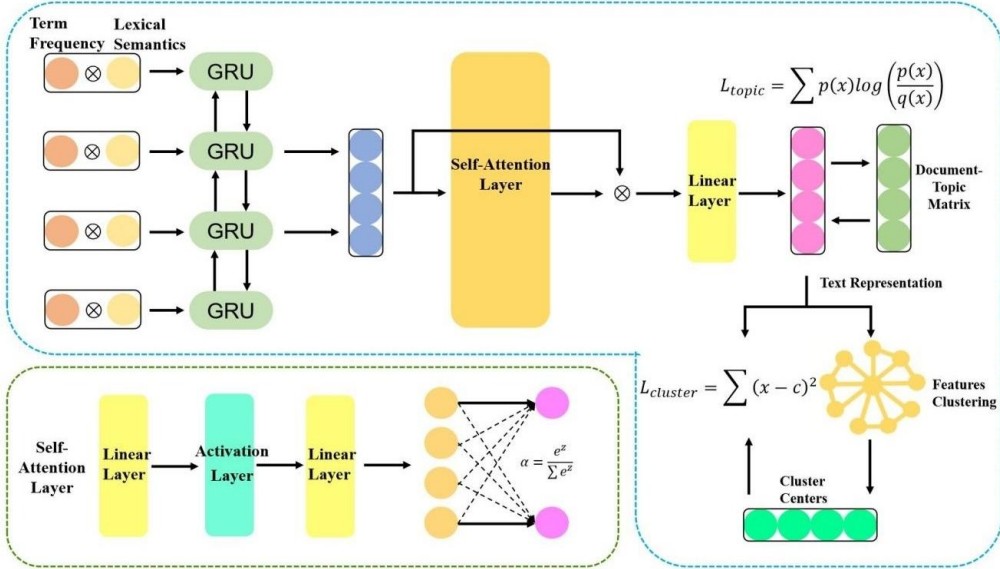

**Figure 1  Double-target self-supervised clustering with multi-feature fusion.**

which were used as the input to the model for deep learning. Meanwhile, a self-attention mechanism was introduced to calculate the weight of each word in the question text, *i.e.,* the interactions between words. To learn fusing cross-document topic features and build an end-to-end text clustering model, we constructed two target functions and integrated them into a unified clustering framework. From Fig. 1, it is the basic structure of our model.

**Text representation module**: We obtained the weight of a word based on term frequency, generated word vectors based on the semantic information, and fused term frequency with word semantic information to obtain weighted word vectors. The document-topic feature was a kind of cross-document topic information and it was trained as a self-supervised target of the model.

**Auto-encoder module**: The text representation was input to the BiGRU model for deep learning of high-quality semantic information, and a self-attention mechanism was introduced to obtain correlations among words in the question text.

**Double-target self-supervision module**: To construct an end-to-end text clustering model and obtain a more friendly clustering representation, the text representation and clustering results were fused under a unified framework, and the clustering target function $L_{cluster}$ was fused to the network loss function of the learning representation. At the same time, the cross-document topic target function $L_{topic}$ was introduced to make the text cover the full document information. The above two functions were combined to construct a joint loss, *i.e.,* $L_{train} = \alpha L_{cluster} + L_{topic}$, achieving double-target self-supervision.

**Table 1  Classification of medical questions.**

| Coding | First-class category | Second-class category | Description |
|---|---|---|---|
| 1 | Getting to know diseases | | Including disease basics and rationale |
| 2 | Disease causes | | Including the causes of diseases and some symptoms |
| 3 | Prevention | | How to prevent diseases |
| 4 | Examination | | Including initial illness examination, routine examination later and relevant knowledge of examination |
| 5 | Diagnosis | | Asking whether have the disease and the possibility of cancer on the basis of symptom description; asking about the pathological type, clinical stage and metastasis of the disease |
| 6 | Treatment | Treatment plan | Describing symptoms and inquiring treatment plan which includes treatment risk, safety and effectiveness. |
| 7 | | Adverse reactions | Adverse reactions or side effects caused by various types of treatment |
| 8 | | Precautions | Precautions before, while and after the treatment |
| 9 | Daily care | Diet | Asking if certain foods could be eaten and what should be paid attention to in diet |
| 10 | | Living habits | Asking if smoking and drinking are allowed and about sleep time |
| 11 | Hospitals and medical insurance | | Asking about hospitals, costs and whether a drug can be reimbursed by medical insurance |
| 12 | Prognosis and survival time | | Asking prognosis, survival time and progression of the illness |

## Collection and pre-processing for medical question texts

The research used Python to obtain the question titles from the medical question and answer community as the research object. Due to the problems of semantic ambiguity and improper language use of the question sentences, the data needed to be pre-processed.

Data cleaning: Meaningless data was removed, and problems such as unusing and misusing punctuation marks were corrected. A few inaccurate and irregular expressions were adjusted. We constructed a medical synonym list for the liver cancer to standardize terms.

Word splitting: Words were split using pkuseg.

Category labeling: The category of question texts is clarified to evaluate clustering effects. After conducting literature research, consulting experts and analyzing the medical questions in medical question and answer community, we divided medical questions into 12 major categories, and the categories are shown in Table 1 below.

## Double-target self-supervised clustering with multi-feature fusion
### Text representation module

Lexical semantic features

The fastText model involves input layer, hidden layer and output layer. The model is developed on the basis of skip-gram, and each word of the input context is decomposed based on word n-gram format. It can also represent the internal order of words. Take the phrase "primary liver cancer" as an example. If the value of n-gram is 2, then its bigram

include "<primary", "primary liver", "liver cancer", and "cancer>". The word vector of "primary liver cancer" can be represented by superimposing the 4 word vectors after decomposition. The lexical semantic features generated by the fastText model contain semantic and partial sequential information. The question text in the medical question-and-answer community has very few words, and the contextual semantics and word order information of the feature word are obviously missing. The lexical semantic features are extracted by the fastText model, and lexical semantic features of the medical question text D are represented as:

$$D = \begin{pmatrix} a_{11} & a_{12} & a_{13} & \cdots & a_{1T} \\ a_{21} & a_{22} & a_{23} & \cdots & a_{2T} \\ \vdots & \vdots & \vdots & \cdots & \vdots \\ a_{k1} & a_{k2} & a_{k3} & \cdots & a_{kT} \end{pmatrix} \tag{1}$$

where $a_k$ denotes the vector of word $w_k$ in the question text D and T is the vector dimension. Weighted lexical semantic features

The TF-IDF values are used to weight the fastText vocabulary semantics, so as to reduce the influence of non-keyword features and improve the ability to distinguish topics. Weight features of the vocabulary in question text D are shown below:

$$W = \left[ tfidf_1, tfidf_2, \ldots, tfidf_k \right] \tag{2}$$

where $tfidf_k$ denotes the weight of the word $w_k$ in the question text D. A bigger TF-TDF value means more importance.

We multiplied the lexical semantic features of the text with the corresponding TF-TDF values, and obtained the weighted lexical semantic features FW of the question text D, with $FW \in \mathbb{R}^{k \times T}$.

$$FW = WF = \begin{pmatrix} a_{11} \times tfidf_1 & a_{12} \times tfidf_1 & a_{13} \times tfidf_1 & \cdots & a_{1T} \times tfidf_1 \\ a_{21} \times tfidf_2 & a_{22} \times tfidf_2 & a_{23} \times tfidf_2 & \cdots & a_{2T} \times tfidf_2 \\ \vdots & \vdots & \vdots & \cdots & \vdots \\ a_{k1} \times tfidf_k & a_{k2} \times tfidf_k & a_{k3} \times tfidf_k & \cdots & a_{kT} \times tfidf_k. \end{pmatrix} \tag{3}$$

### Auto-encoder module

BiGRU

The encoder is essentially a RNN that encodes the input sequence into a feature representation. For lexical sequence prediction, an arbitrary question text sequence $X = (x_1, x_2, \ldots, x_k)$, $x_i \in \mathbb{R}^T$ is given, where k is the number of words in the question text and T is the dimension of the pre-trained word embedding, *i.e.,* the topic dimension. Then, the encoder was used to learn the mapping $h_t = f_1(h_{t-1}, x_t)$ from $x_t$ to $h_t$ at time step t, where $h_t \in \mathbb{R}^m$ is the hidden state of the encoder at time t, m is the size of the hidden state, and $f_1$ is a nonlinear function. In the study, BiGRU was used as $f_1$ to obtain long-term dependent neuron information. Another reason for using GRU units was that the neurons sum over a certain time, which helps to overcome the problem of gradient disappearance

and better capture long-term correlations in time series. Each GRU unit consists of two Sigmoid gates, namely the reset gate $r_t$ and the update gate $z_t$ . The GRU unit is updated as follows:

$$r_t = \sigma(W_r[h_{t-1}, x_t]) \tag{4}$$

$$z_t = \sigma(W_z[h_{t-1}, x_t]) \tag{5}$$

$$\tilde{h}_t = \tanh(W_{\tilde{h}}[r_t h_{t-1}, x_t]) \tag{6}$$

$$h_t = (1 - z_t)h_{t-1} + z_t h_{\tilde{t}} \tag{7}$$

$$y_t = \sigma(W_o h_t) \tag{8}$$

where $[h_{t-1}, x_t] \in \mathbb{R}^{(m+T)}$ is the connection of the previous hidden state $h_{t-1}$ and the current input $x_{t..}$; $x_t \cdot W_r, W_z, W_{\tilde{h}} \in R^{m \times (m+T)}$ are parameters to be learned; $\sigma$ is the sigmoid function. A bidirectional structure was used to obtain the dependencies between adjacent words in a single question text.

$$\overrightarrow{h_t} = \overrightarrow{GRU}(W_t, \overrightarrow{h_{t-1}}) \tag{9}$$

$$\overleftarrow{h_t} = \overleftarrow{GRU}(W_t, \overleftarrow{h_{t-1}}) \tag{10}$$

$$h_t = [\overrightarrow{h_t}; \overleftarrow{h}_t]. \tag{11}$$

Finally $\overrightarrow{h_t}$ and $\overleftarrow{h}_t$ are horizontally spliced to obtain a hidden state $h_{t..}$ Let the number of hidden units per one-way GRU be u. For simplicity, all $h_t$ is $H \in \mathbb{R}^{k \times 2u}$, where k is the number of words in the input question text.

Self-attention mechanism

The self-attention mechanism is a variation of the attention mechanism. Unlike the attention mechanism, the encoder and decoder of the self-attention mechanism deal with the same text. According to the word distribution in the text, the mechanism calculates the weight of each word, *i.e.,* the interaction between words, which reduces the reliance on external information and is better at capturing the internal relevance of data or features. Each feature was assigned different weights, with complementary feature information enhanced and the conflicting parts weakened. A common approach in many previous studies has been to use the final hidden state of the RNN or to establish the simple vector representation through the max-pooling (or mean-pooling) of RNN hidden state. However, it is relatively difficult to have multiple semantics in all time steps of an RNN model and there are lexical semantics that are not necessary. Different from previous approaches, a self-attention mechanism was introduced which allowed different aspects of a sentence to be extracted into multiple vector representations. In the proposed sentence embedding model, a self-attention mechanism was introduced in BiGRU to obtain more

information without other additional input. It relieved the encoder of some long-term memory burden due to the direct access to the hidden representation at all times. The computational procedure of the self-attention layer is as follows:

$$Z = W_2 \cdot tanh\left(W_1 \cdot H^T\right) \tag{12}$$

$$A = \frac{\exp(Z)}{\sum \exp(Z)} \tag{13}$$

$$\acute{Z} = A \cdot H^T \tag{14}$$

where $W_1$ and $W_2$ are linear layers with learnable parameters; tanh is the activation layer with nonlinear transformation; H is the output feature of GRU; Z is the linear layer output feature. The self-attention matrix A was computed by the Softmax function. Exp represents the exponential function, and each value in A represents the correlation for the feature value with other feature values. The enhanced intermediate feature $\acute{Z}$ on the basis of the self-attention mechanism was obtained by multiplying the attention matrix A with the input H. Then the output features are obtained through the linear projection layer calculation of $\acute{Z}$, which is illustrated as follows:

$$M = W_2 tanh\left(W_1^T \acute{Z}\right) \tag{15}$$

where $W_1$ and $W_2$ are linear layers with learnable parameters; tanh is the activation layer with nonlinear transformation. $M \in R^{1 \times r}$ is the final text representation vector, where $r$ is the vector dimension.

### Double-target auxiliary module

To construct a model of end-to-end feature extraction text clustering and incorporate cross-document topic information to obtain a more friendly clustering representation, we introduced the target loss function $L_{topic}$ for cross-document topic information, fused the clustering target loss function $L_{cluster}$ into the network loss function of the learning representation, and constructed the joint loss function in a unified framework. In this way, the text representation was fused with clustering results under a unified framework.

**The cross-document topic target loss function $L_{topic}$.** To further learn cross-document-topic information, the document-topic distribution T was used as the self-supervised training target. Under the assumption that there was one-to-one mapping relationship between document clusters and topics, it enabled the auto-encoder BiGRU to learn a relatively clustering-friendly representation with the cross-document information in the training phase. However, it was necessary to obtain a prediction vector with the same dimensionality as the corresponding document-topic vector. We used a two-layer learnable fully-connected layer to reduce the dimensionality of the text representation vector M.

The Softmax function was used to calculate the posterior probability distribution of the text representation $M$, as illustrated in the following formula.

$$D = Soft \max(W_2 Relu(W_1^T M)) \tag{16}$$

where $D \in R^{1 \times r}$; $r$ is the length of the document-topic distribution vector; ReLU is a linearly corrected activation function that serves to set input negative values to zero; $W_1$ and $W_2$ are linear layers with learnable parameters for the dimensionality reduction of the text representation vector $M$, which make the dimensionality of $M$ the same as that of the document-topic vector.

The result $d_{ij} \in D$ represents the probability that the sample i belongs to the topic j and D is a probability distribution. The final loss function can be defined as follows:

$$L_{\text{topic}} = KL(T||D) = \sum_i \sum_j t_{ij} \log \frac{t_{ij}}{d_{ij}}. \qquad (17)$$

T is the document-topic distribution of the text computed by the topic model LDA, *i.e.,* the probability distribution of each sentence related to the topic. The KL-divergence was used as the training model of the loss function.

**The clustering target loss function L_cluster.** To construct end-to-end text clustering models and enable auto-encoder BiGRU to learn a friendly clustering representation, we proposed a batch-based clustering loss to improve the clustering of output features, which was inspired by the research of *Xie, Girshick & Farhadi (2016)* and *Wen et al. (2016)*. This loss is represented as follows:

$$L_{cluster} = \sum_{i=1}^{m} \sum_{k=1}^{K} (x_{i,k} - c_k)^2 \qquad (18)$$

where m is the number of samples within each batch; K is the number of clustering centroids within each batch, *i.e.,* the number of clustering categories 12; c is the clustering centroid of each batch; $x$ is the samples within each batch. In each training iteration, the clustering centroids c was obtained by clustering the features within the batch through the K-Means algorithm. $L_{\text{cluster}}$ made the features within the batch close to the clustering centroid c.

**The joint target loss function L_train.** Combining the two target functions designed, we obtained the overall loss of the training phase, which is shown as follows.

$$L_{\text{train}} = \alpha L_{cluster} + L_{topic} \qquad (19)$$

where $\alpha$ is the hyperparameter. The loss $L_{\text{train}}$ was continuously being optimized during the clustering, updating the model parameters. In the early iterations of the algorithm, the features extracted by the model were not yet ideal, and the $L_{\text{cluster}}$ with a too large $\alpha$ might produce side effects. Hence we used a simple climbing strategy to adjust $\alpha$, so that it gradually increased with the training iterations proceeding, as shown in the following formula:

$$\alpha(t) = \frac{t}{T} * \alpha_{max}, t \leq T' \qquad (20)$$

where $\alpha_{\text{max}} = 0.3$; t is the number of iterations; and T' is the fixed number of iterations.

# EXPERIMENT AND ANALYSIS

## Experimental environment and evaluation indicators

The programming language used for the experiment was Python 3.8, the integrated development environment (IDE) was Pytorch 1.9.0, and the experimental environment was a workstation with Ubuntu 16.04 operating system. The workstation has 8 NVIDIA TITAN RTX 24 GB GPUs, 256G RAM and 80 Intel® Xeon® Gold 6248 CPUs @ 2.50 GHz. The AdamW optimizer was used in the experiment. The learning rate was set $1e-3$, and the learning rate strategy was linear decay. 32 epochs were trained using a batch size of 256.

The BiGRU used a hidden state dimension of size 512, the linear projection layer used an output dimension of 64, and the output text vector is 64 in length. Network parameters were generated by random normal distribution and updated through stochastic gradient descent (SGD).

The indicators for evaluating the clustering effect in this experiment are Average Clustering Accuracy (ACC) (*McDaid, Greene & Hurley, 2011*) Normalized Mutual Information (NMI) (*Kabir, Shahjahan & Murase, 2012*), Adjusted Rand Indicator (ARI) (*Yang et al., 2010*) and F1 (*Shi et al., 2020*). ACC is used to evaluate the accuracy of the prediction of the experimental category compared with the real category; NMI and F1 are used to predict the similarity between the experimental category and the real category; ARI measures the degree of coincidence of the two data distributions.

## Comparative experimental design

Six comparative experiments were designed for this study, as shown in follows:

Comparison method 1 (M1): We used a clustering framework based on the K-Means algorithm, and implemented a spatial vector model and a TF-IDF feature weighting scheme for text representation (*Soares et al., 2019*). The method was baseline.

Comparison method 2 (M2): We used a clustering framework based on K-Means algorithm, and used BERT model for text representation.

Comparison method 3 (M3): We used TAE model (*Jin, Zhao & Ji, 2020*) which optimizes the text representation through combining BoW methods and neural sentence embedding approaches, and proposed a self-supervised method with document-topic information distribution as the target.

Comparison method 4 (M4): SIF-Auto (*Hadifar et al., 2019*) was used. It is an algorithm that uses SIF word vectors for text representation, extracts features through self-encoding networks, and regards clustering distribution as an auxiliary target distribution for self-supervision.

Comparison method 5 (M5): This research introduced the text representation method, BERT_AE_K-Means (*Zhu et al., 2022*). It used the pre-trained model BERT to extract text semantic representation, extracted features by auto-encoders, and used clustering targets as auxiliary target distribution for self-supervision.

**Table 2  Experimental clustering effects of different models.**

| Model | Model | NMI | ARI | ACC | F1 |
|-------|-------|------|------|------|------|
| M1 | baseline | 0.3472 | 0.3746 | 0.8376 | 0.4705 |
| M2 | BERT | 0.2021 | 0.1565 | 0.8005 | 0.2694 |
|    | Pre-BERT | 0.3704 | 0.3384 | 0.8291 | 0.4391 |
| M3 | TAE | 0.4002 | 0.4870 | 0.8648 | 0.5671 |
| M4 | SIF-Auto | 0.1726 | 0.0517 | 0.6474 | 0.2491 |
| M5 | Pre-BERT_AE | 0.3898 | 0.3440 | 0.8306 | 0.4438 |
| M6 | MF-DSC | 0.4346 | 0.4934 | 0.8649 | 0.5737 |

## Analysis of comparative experiment results

According to the above comparative experiment design, five comparative experiments were done on the constructed medical question dataset, and the clustering effect of each model is shown in Table 2. The MF-DSC text clustering method outperforms the other models with 0.4346, 0.4934, 0.8649 and 0.5737 for NMI, ARI, ACC and F1, respectively. Compared with the traditional clustering methods of spatial vector model M1 and pre-training model M2 for text representation, MF-DSC performs much better in each indicator. The spatial vector model increases the text vector dimensionality and ignores the semantic relationship among words when dealing with large-scale text. The pre-training model BERT generates high-dimensional data, and as s result, the semantic information of the question text cannot be better represented. In the pre-training models M2, Pre-BERT and BERT are conducted respectively. Pre-BERT uses the knowledge mask strategy for pre-training and BERT does not. As a result, the clustering effect of pre-trained Pre-BERT is greatly improved, which can improve the representation for texts, reduce the difference of data feature distribution between the pre-trained corpus and the target corpus, and get the dynamic word vector more appropriate for the semantic environment. M3, M4, and M5 are the models proposed in recent years that perform well on short text clustering tasks. Compared with M3, MF-DSC performs better in all four indicators, with an increase of 3.44 percentage points in NMI. Both MF-DSC and M3 use cross-document topic information as the self-supervised target. However, MF-DSC also fuses the clustering target function into the clustering framework for a more friendly clustering representation. Compared with M4 and M5, MF-DSC performs much better in all four indicators. In M4, SIF weights average of the lexical vectors of all words and then removes the common part. Since the question text is short with sparse features, M4 brings more information loss, making it the least satisfactory among all models. M5 uses the auto-encoder, on the basis of the pre-training model, to extract features and reduce dimensionality, with effect better than that of Pre-BERT in M2.

To represent the changes of clustering results more clearly, this study used the t-SNE algorithm to downscale the clustering results for medical question texts from feature space to 2-dimensional space. The visualization results under the plane coordinate system are shown in Fig. 2, where the sample data are divided into 12 categories, and different colors represent different categories. From the figure, the M1 baseline of Fig. 2A based on the traditional spatial vector model shows the worst effect, for it does not effectively divide

the data of each category. This is because the question text has a small vocabulary and few core words, and the text representation simply weighted by TF-IDF features cannot fully characterize the core word semantics. In Fig. 2B, the BoW model and the word embedding are combined to optimize the text representation, including lexical topic features and lexical semantic features. The boundaries of each cluster are more clearly delineated. In Fig. 2C, an auto-encoder is used to extract high-order features from the data and reduce the dimensionality. A clustering objective optimization function is constructed, improving the clustering effect. From Fig. 2D, the text clustering method MF-DSC divides the text data more clearly. It fuses term frequency, lexical semantic and cross-document feature information, and unites cross-document topic targets and clustering targets. Hence the boundaries of each cluster are clearer and the outlier and intersection points become less, producing better clustering effect and improving the effectiveness of clustering.

## Analysis of ablation experiments

To verify the role played by term frequency features, semantic features, and the two auxiliary target functions on the algorithm, this study conducted ablation experiments on MF-DSC, shown in Table 3. Firstly, in terms of feature ablation, the absence of term frequency features decreased 0.0331, 0.0111, 0.0017 and 0.0117 points in NMI, ARI, ACC and F1, respectively; the absence of lexical semantic features decreased 0.0146, 0.0212, 0.0079 and 0.0164 points. The clustering indicator values were reduced after removing either term frequency features or lexical semantic features. The term frequency affects more the similarity degree between the clustered categories and the true categories, and lexical semantic features more the degree of coincidence between the two data distributions. In terms of auxiliary target ablation, 0.0393, 0.0168, 0.0031 and 0.0152 points were reduced on the four indicators respectively without $L_{topic}$; the reduction values were 0.0128, 0.0256, 0.0087 and 0.0203 on the four indicators respectively if there was no $L_{cluster}$. The absence of either $L_{topic}$ or $L_{cluster}$ affects the text clustering effect, and since $L_{topic}$ enables the deep learning network to learn cross-document information, it has a greater impact on the similarity between clustered categories and real categories.

## Model generalization analysis

The performance improvement of MF-DSC on the medical question dataset has been verified. To make the conclusions more reliable, this study conducted experiments on other six text datasets. These six text datasets are Twitter US Airline Sentiment, 20 Newsgroups, online shopping 10 cats, BDCI2018, Illness-dataset, and AGNews.

Twitter US Airline Sentiment: This dataset is a record of comments about US airlines on Twitter and is often used for sentiment identification. It includes the ID of tweets, the sentiment of tweets (including three categories, positive, negative, and neutral), reasons for negative tweets, the name of the airline and tweet texts.

20 Newsgroups: This dataset is a collection of approximately 20,000 newsgroup documents partitioned (nearly) evenly across 20 different newsgroups.

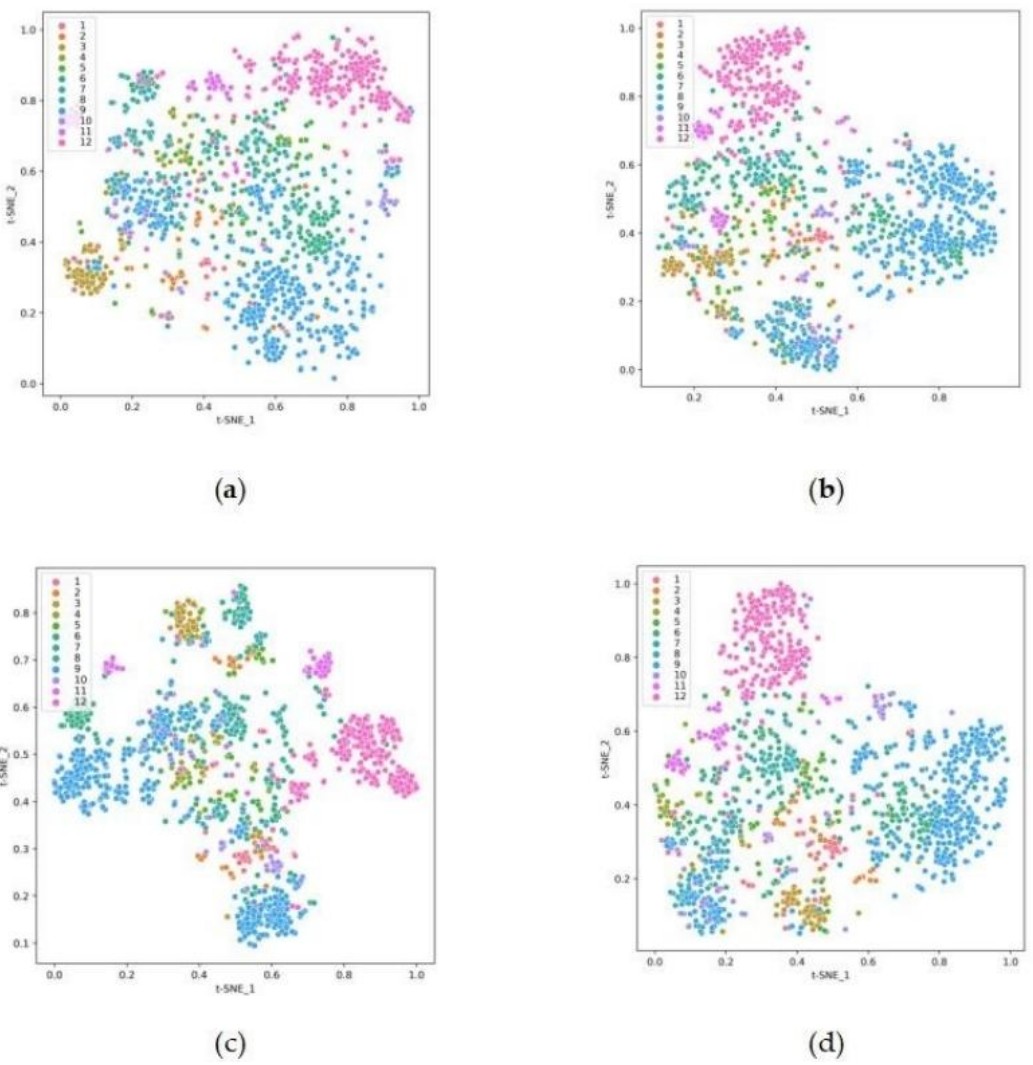

**Figure 2** Visualization of text clustering results for medical question texts (A) baseline, (B) TAE, (C) Pre-Bert-AE, (D) MF-DSC.

Online shopping 10 cats: This dataset comes from the product reviews on e-commerce websites, and involves 10 categories, *i.e.,* books, tablets, cell phones, fruits, shampoos, water heaters, Mengniu dairy products, clothes, computers and hotels.

BDCI2018: This dataset is from user review data on automotive forums and contains 10 categories: power, price, interior trim, configuration, safety, appearance, handling, fuel consumption, space, and comfort.

Illness-dataset: The dataset consists of 22,660 tweets collected in 2018 and 2019. It spans across four domains: Alzheimer's, Parkinson's, Cancer, and Diabetes.

AGNews: The news articles of this dataset have been gathered from more than 2,000 news sources by ComeToMyHead, an academic news search engine. The dataset has a total of 120,000 training samples and 7,600 test samples. There are four tags, but no specific

**Table 3  Model ablation experiments.**

| Model | NMI | ARI | ACC | F1 |
|---|---|---|---|---|
| MF-DSC | 0.4346 | 0.4934 | 0.8649 | 0.5737 |
| Feature ablation | | | | |
| MF-DSC (without term frequency) | 0.4015 | 0.4823 | 0.8652 | 0.5620 |
| MF-DSC (without lexical semantic features) | 0.4200 | 0.4722 | 0.8570 | 0.5573 |
| auxiliary target ablation | | | | |
| MF-DSC (without $L_{topic}$) | 0.3953 | 0.4766 | 0.8618 | 0.5585 |
| MF-DSC (without $L_{cluster}$) | 0.4218 | 0.4678 | 0.8562 | 0.5534 |

meaning of the tag is given. The text contains the news headlines as well as the body text, and only the news headlines are used in this study.

The details of each dataset, including the training set, test set, category number, average text length, maximum text length, minimum text length, and language, are shown in Table 4.

Table 5 shows the NMI, ARI, ACC and F1 values of the three comparison methods and MF-DSC tested on the six public datasets. Except that the MF-DSC text clustering method is slightly inferior to the Baseline model in terms of ACC on the Illness-dataset, the MF-DSC outperforms other models on other datasets.

The comparison of NMI values in every dataset of different models is shown in Fig. 3. Among the four models, MF-DSC performs best in all six datasets. The highest NMI appears in Online shopping 10 cats, with a value of 0.4319, followed by 20 Newsgroups, BDCI2018, and AGNews in sequence. The NMI value is lower in Twitter US Airline Sentiment and Illness-dataset. Online shopping 10 cats is a product reviews dataset in Chinese, where MF-DSC shows a better generalization effect. 20 Newsgroups is a long text in English with 20 categories, and BDCI2018 is a short text in Chinese with 10 categories. The NMI performance of 20 Newsgroups and BDCI2018 is at the same level, while the NMI in Twitter US Airline Sentiment, Illness-dataset and AGNews is relatively low, which may be related to small number of categories and unclear classification in the dataset.

In summary, MF-DSC fuses term frequency features, lexical semantic features, and cross-document features to obtain deep semantics, and constructs an optimized end-to-end clustering model combining cross-document topic targets and clustering targets to obtain a more friendly clustering representation, showing better clustering effects. Meanwhile, it has been proved effective in other six datasets in spite of the medical question dataset.

## CONCLUSION

In the current field of natural language processing, text clustering, as an unsupervised learning method, plays a crucial role in tasks such as information extraction, topic analysis, and intelligent question answering. Especially in the field of medicine, text clustering can help researchers quickly discover similar research topics or disease information from massive medical literature, thereby accelerating the progress of medical research.

**Table 4  Details of text datasets.**

| Dataset | Training set | Test set | Category number | Average text length | Maximum text length | Minimum text length | Language |
|---|---|---|---|---|---|---|---|
| Twitter US Airline Sentiment | 12,640 | 2,000 | 3 | 103.82 | 186 | 12 | en |
| 20 Newsgroups | 11,314 | 7,532 | 20 | 1,823.82 | 160,456 | 112 | en |
| Online shopping 10 cats | 54,773 | 8,000 | 10 | 58.41 | 2,876 | 1 | zh |
| BDCI2018 | 7,949 | 2,000 | 10 | 49.21 | 200 | 10 | zh |
| Illness-dataset | 18,660 | 4,000 | 4 | 162.05 | 339 | 19 | en |
| AGNews | 120,000 | 7,600 | 4 | 43.06 | 116 | 7 | en |

**Table 5  Experimental results of model generalization.**

| Metric | | Model | | | |
|---|---|---|---|---|---|
| | | Baseline | Pre-Bert-AE | TAE | MF-DSC |
| Twitter US Airline Sentiment | NMI | 0.0514 | 0.0582 | 0.0601 | 0.0664 |
| | ARI | 0.0805 | 0.0702 | 0.0734 | 0.0886 |
| | ACC | 0.5444 | 0.5381 | 0.5392 | 0.5469 |
| | F1 | 0.4929 | 0.4977 | 0.5021 | 0.5093 |
| 20 Newsgroups | NMI | 0.1864 | 0.2281 | 0.1928 | 0.2243 |
| | ARI | 0.0829 | 0.0935 | 0.0919 | 0.0977 |
| | ACC | 0.8971 | 0.8939 | 0.9041 | 0.8974 |
| | F1 | 0.1362 | 0.1478 | 0.1422 | 0.1507 |
| online shopping 10 cats | NMI | 0.2895 | 0.3377 | 0.3702 | 0.4319 |
| | ARI | 0.1451 | 0.1922 | 0.2471 | 0.2897 |
| | ACC | 0.8057 | 0.8115 | 0.8306 | 0.8364 |
| | F1 | 0.2565 | 0.3011 | 0.3439 | 0.3839 |
| BDCI2018 | NMI | 0.0145 | 0.1165 | 0.2121 | 0.2139 |
| | ARI | 0.0048 | 0.0357 | 0.0891 | 0.0958 |
| | ACC | 0.7597 | 0.7775 | 0.7737 | 0.7788 |
| | F1 | 0.1446 | 0.1633 | 0.2214 | 0.2248 |
| Illness-dataset | NMI | 0.0264 | 0.0321 | 0.0455 | 0.0456 |
| | ARI | 0.0237 | 0.0368 | 0.0396 | 0.0397 |
| | ACC | 0.6249 | 0.6052 | 0.6128 | 0.6128 |
| | F1 | 0.2769 | 0.3097 | 0.3062 | 0.3062 |
| AGNews | NMI | 0.0481 | 0.0491 | 0.1011 | 0.1197 |
| | ARI | 0.0332 | 0.0334 | 0.0584 | 0.0834 |
| | ACC | 0.6004 | 0.6013 | 0.5993 | 0.6007 |
| | F1 | 0.2758 | 0.2759 | 0.2794 | 0.2809 |

However, medical problem texts often have the characteristics of few characters and sparse features, which makes traditional text clustering algorithms difficult to accurately capture the deep semantic relationships of the text. In response to this issue, this paper proposes a novel medical problem text clustering method–the MF-DSC model. This model achieves deep semantic representation of medical problem texts by integrating term

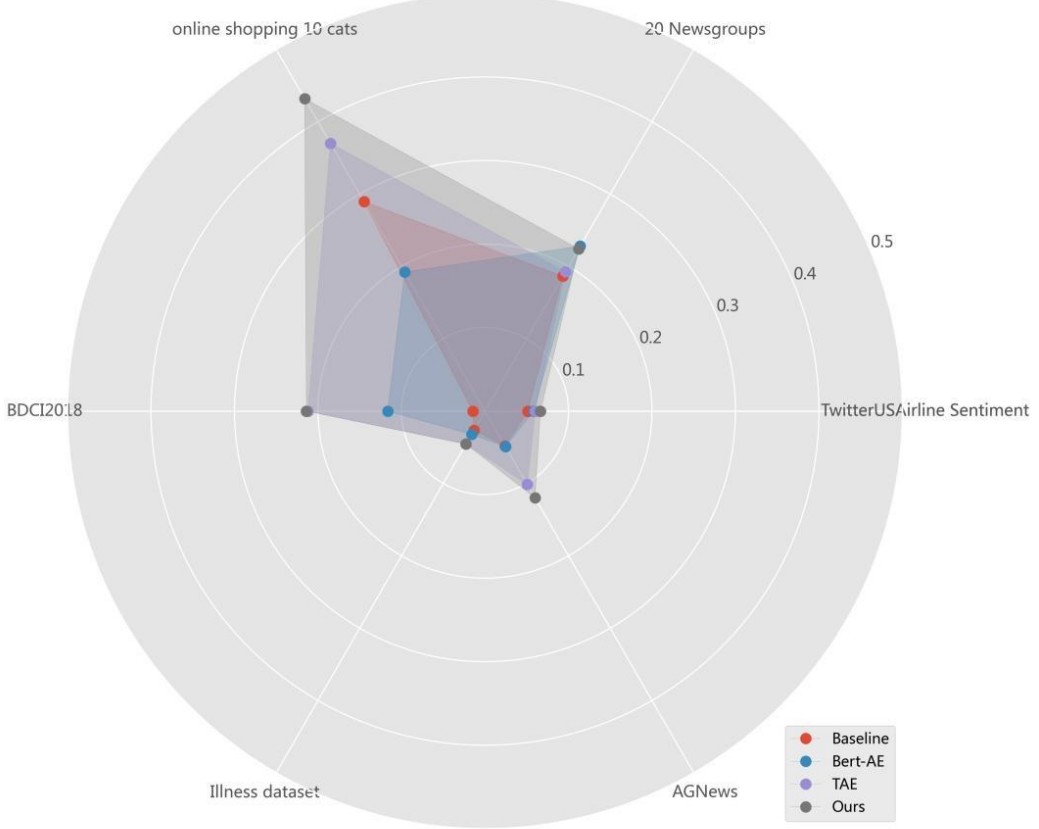

**Figure 3** **NMI values for different models in every dataset.**

frequency information and lexical semantic information, achieving significant performance improvement.

The overall framework of the MF-DSC model includes a text representation module, an encoder module, and a dual object assistance module. In the text representation module, we adopt the method of weighted word vectors to integrate term frequency information and lexical semantic information into the word vectors, thereby achieving deep semantic representation of the text. Specifically, we calculate the frequency of the term in the text and use it as a weight to sum it with the pre trained word vector to obtain the representation vector for each text. In this way, we can not only capture the frequency information of terms, but also fully utilize the semantic information of vocabulary, making the text representation more accurate and comprehensive.

In the encoder module, we adopt the encoder structure of deep learning, taking the text representation vector as input and obtaining the deep feature representation of the text through nonlinear transformation of multi-layer neural networks. The encoder structure can effectively capture deep semantic relationships in text, making clustering results more accurate and reliable. In order to further improve the performance of clustering, we introduced a dual objective auxiliary module in the MF-DSC model. This

module includes cross document topic information objectives and clustering objectives. By optimizing these two objective functions, we can guide the model to learn more accurate text representations and clustering structures. This end-to-end clustering method enables the MF-DSC model to have stronger adaptability and generalization ability in medical problem text clustering tasks. To verify the effectiveness of the MF-DSC model, we conducted experiments on a medical problem text dataset. The experimental results show that the MF-DSC model outperforms the baseline model and the other four comparative models in evaluation indicators such as NMI, ARI, ACC, and F1. This fully demonstrates the superior performance of the MF-DSC model in medical problem text clustering tasks. In order to further explore the roles of various components in the MF-DSC model, we also conducted ablation experiments. The experimental results indicate that item frequency, lexical semantics, and two auxiliary objective functions all play an important role in the performance of the algorithm. This further confirms the rationality and effectiveness of the MF-DSC model design.

In addition, to verify the universality of the MF-DSC model, we also conducted experiments on six publicly available datasets. The experimental results show that the MF-DSC model has achieved excellent performance in text clustering tasks of different fields and scales. This fully demonstrates the universality and robustness of the MF-DSC model.

### Funding
This research was supported by the National Key Research and Development Program of China (2020AAA0104905). The funders had no role in study design, data collection and analysis, decision to publish, or preparation of the manuscript.

### Grant Disclosures
The following grant information was disclosed by the authors:
The National Key Research and Development Program of China: 2020AAA0104905.

### Competing Interests
The authors declare there are no competing interests.

### Author Contributions
- Xifeng Shen conceived and designed the experiments, performed the experiments, analyzed the data, performed the computation work, prepared figures and/or tables, authored or reviewed drafts of the article, and approved the final draft.
- Yuanyuan Sun analyzed the data, authored or reviewed drafts of the article, formal analysis and investigation, and approved the final draft.
- Chunxia Zhang analyzed the data, authored or reviewed drafts of the article, and approved the final draft.
- Cheng Yang analyzed the data, authored or reviewed drafts of the article, and approved the final draft.

- Yi Qin analyzed the data, authored or reviewed drafts of the article, project administration, and approved the final draft.
- Weining Zhang performed the experiments, analyzed the data, prepared figures and/or tables, and approved the final draft.
- Jiale Nan performed the experiments, analyzed the data, prepared figures and/or tables, and approved the final draft.
- Meiling Che performed the experiments, analyzed the data, prepared figures and/or tables, and approved the final draft.
- Dongping Gao conceived and designed the experiments, authored or reviewed drafts of the article, and approved the final draft.

## Data Availability

The code and raw data are available in the Supplemental Files.

## Supplemental Information

Supplemental information for this article can be found online at http://dx.doi.org/10.7717/peerj-cs.2075#supplemental-information.

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
