# Peer review of "Double-target self-supervised clustering with multi-feature fusion for medical question texts"

_PeerJ Computer Science, doi:10.7717/peerj-cs.2075_

## Round 0.1 · original submission · Minor Revisions

This paper proposes a novel approach called MF-DSC for clustering medical question texts. The key contributions include 1) fusing term frequency and lexical-semantic features to obtain more informative text representations, 2) using a self-attention mechanism to capture word interactions, and 3) integrating cross-document topic modelling and clustering objectives in a unified end-to-end clustering framework. Experiments on datasets demonstrate the effectiveness of the proposed MF-DSC method compared to several baselines.

Pros:
1. Addressing the challenges of short text clustering for medical questions is an important problem with practical significance. The paper provides good motivation.
2. Combining multiple feature types (term frequency and lexical semantics) is a strength that enhances the text representations compared to using a single feature type.
3. Introducing a self-attention mechanism to model word interactions is a novel aspect that can capture a richer context.
4. Unifying the cross-document topic modelling objective and clustering objective in an end-to-end framework is an elegant approach to learning clustering-friendly representations.
5. The extensive experiments on multiple datasets, ablation studies, and generalization analysis provide a comprehensive evaluation.


Cons:
1. Provide more details on the experimental setup, model architectures, and hyperparameter settings.
2. As reviewers' suggestions, the authors should proofread the paper carefully and fix grammatical mistakes and phrasing to enhance the presentation.
3. Expand the related work section with a deeper literature review.

**Language Note:** The Academic Editor has identified that the English language must be improved. PeerJ can provide language editing services - please contact us at [email protected] for pricing (be sure to provide your manuscript number and title). Alternatively, you should make your own arrangements to improve the language quality and provide details in your response letter. – PeerJ Staff

Reviewer 1 ·

Basic reporting

This paper solves the problem that the features extracted from a single model cannot fully represent the text content, and proposes an important model - a double target self supervised clustering with multi feature fusion (MF-DSC).

Experimental design

The experimental results show that compared with the traditional single model, this model can better handle unstructured text, especially medical question and answer data with short characters and irregular language use.

Validity of the findings

Overall, this article is well organized and presented well.

Additional comments

However, there are still some minor issues that need improvement:
(1) The author should summarize the main contributions of this article in Section 1.
(2) Suggest discussing the limitations of this work at the end of the article.
(3) There are some typing and grammar errors in this article.

Reviewer 2 ·

Basic reporting

This study proposes a multi-feature fusion-based double-target self-supervised clustering method (MF-DSC) for processing unstructured texts related to medical fields. The overall structure of the article is scientific and reasonable, and the logic is clear.

Experimental design

The authors conducted extensive experiments to validate the paper's claims. Compared with the other 5 models to evaluate the effectiveness of MF-DSC, the results show that MF-DSC outperformed baseline models in various metrics.

Validity of the findings

It can be seen that compared with the traditional single model, MF-DSC shows significant advantages in processing unstructured text, especially medical question-and-answer data with short characters and non-standard language use.

Additional comments

In Table 2, there are two columns "Model". it would be better to remove the first column.
There are some typing and grammar errors in the article. Some conceptual expressions were slightly vague. Some citations in the article were not properly annotated and should be added following related work.

@article{wei2023medical,
title={Medical question summarization with entity-driven contrastive learning},
author={Wei, Sibo and Lu, Wenpeng and Peng, Xueping and Wang, Shoujin and Wang, Yi-Fei and Zhang, Weiyu},
journal={arXiv preprint arXiv:2304.07437},
year={2023}
}

Reviewer 3 ·

Basic reporting

no comment

Experimental design

no comment

Validity of the findings

no comment

Additional comments

The new model proposed in this article - a double-target self-supervised clustering with multi-feature fusion (MF-DSC) has good processing ability for unstructured text. Please confirm if the citation format of the literature is correct, complete, and in accordance with the prescribed requirements. It is recommended to annotate charts according to journal requirements. For the final conclusion of the article, it is suggested to add some discussion on limitations.

---

## Round 0.2 · accepted · Accept

The reviewers think that the author's responses address their concerns, and now I recommending to accept this paper.

Reviewer 1 ·

Basic reporting

No commet.

Experimental design

No commet.

Validity of the findings

No commet.

Additional comments

This is a second round reviewed paper. The author has revised the questions raised by the last reviewer.

Reviewer 2 ·

Basic reporting

The revised version has addressed my concerns and comments.

Experimental design

no comment

Validity of the findings

no comment